# A cytomegalovirus inflammasome inhibitor reduces proinflammatory cytokine release and pyroptosis

Yingqi Deng [1], Eleonore Ostermann [1] & Wolfram Brune [1] ✉

In response to viral infection, cells can initiate programmed cell death (PCD), leading to a reduction in the release of viral progeny. Viruses have therefore evolved specific mechanisms to curb PCD. Cytomegaloviruses (CMVs) are sophisticated manipulators of cellular defenses and encode potent inhibitors of apoptosis and necroptosis. However, a CMV inhibitor of pyroptosis has not been clearly identified and characterized. Here we identify the mouse cytomegalovirus M84 protein as an inhibitor of pyroptosis and proinflammatory cytokine release. M84 interacts with the pyrin domain of AIM2 and ASC to inhibit inflammasome assembly. It thereby prevents Caspase-1-mediated activation of interleukin 1β (IL-1β), IL-18, and Gasdermin D. Growth attenuation of an M84-deficient MCMV in macrophages is rescued by knockout of either *Aim2* or *Asc* or by treatment with a Caspase-1 inhibitor, and its attenuation in infected mice is partially rescued in *Asc* knockout mice. Thus, viral inhibition of the inflammasome-pyroptosis pathway is important to promote viral replication in vivo.

Viral infection initiates host antiviral responses, which include the release of inflammatory cytokines and the induction of PCD[1,2]. Apoptosis, necroptosis, and pyroptosis are three major forms of PCD that are activated by different sensors and engage specific signaling pathways leading to cell death. While apoptotic cell death is associated with little or no inflammation, necroptosis and pyroptosis are pro-inflammatory: they involve the loss of membrane integrity and the release of inflammatory mediators into the extracellular space[3]. In recent years, it has become increasingly clear that the three cell death modalities do not exist separately, but are closely intertwined. In some cases, a multiprotein signaling complex may activate all three forms simultaneously. This combined cell death mode has been named PANoptosis because it combines characteristics of pyroptosis, apoptosis, and necroptosis[4].

As obligate intracellular pathogens, viruses depend on the metabolism of the cell. Premature demise of infected host cells can profoundly limit viral replication and spread. Therefore, many viruses have developed countermeasures to prevent or delay programmed cell death[1,2]. Human cytomegalovirus (HCMV), a medically significant pathogen[5], and mouse cytomegalovirus (MCMV), a related mouse pathogen used to study viral pathogenesis in vivo[6], are prototypic β-herpesviruses with large genomes and protracted replication cycles. As such, they should be particularly vulnerable to cellular defense mechanisms such as PCD. Not surprisingly, the CMVs express potent inhibitors of apoptosis[7–10] and necroptosis[11–13]. However, a CMV-encoded inhibitor of pyroptosis has not been described, yet.

The inflammasome is a multiprotein complex responsible for the activation of the inflammatory cytokines interleukin 1β (IL-1β) and IL-18[14]. It consists of a sensor, the adaptor protein apoptosis-associated speck-like molecule containing a caspase recruitment domain (ASC or PYCARD), and Pro-Caspase-1, the inactive precursor protein of the cysteine protease Caspase-1. Activation of inflammasome-associated sensors triggers inflammasome assembly and autocatalytic activation of Caspase-1. The active protease cleaves the cytokine precursors pro-IL-1β and pro-IL-18 leading to the release of the mature cytokines. Caspase-1 also activates Gasdermin D (GSDMD), a pore-forming protein[15]. Disruption of plasma membrane integrity by GSDMD pores facilitates the release of IL-1β and IL-18 and ultimately leads to pyroptosis[16]. Sensors involved in inflammasomes comprise NLRs (nucleotide-binding oligomerization domain and leucine-rich repeat-

[1]Leibniz Institute of Virology (LIV), Hamburg, Germany. ✉e-mail: wolfram.brune@leibniz-liv.de

containing receptors) and ALRs (AIM2-like receptors). While the different NLRs respond to a wide variety of pathogen or danger-associated molecular patterns, AIM2 (absent in melanoma 2) and related receptors are activated by dsDNA of viral or bacterial origin[17].

The AIM2 inflammasome is expressed in cells of the myeloid lineage such as macrophages and dendritic cells[18]. Initial studies have demonstrated its activation by vaccinia virus and MCMV[19,20], and later studies have shown its activation by other DNA viruses (reviewed in[21]). By contrast, little is known about viral interference with AIM2 inflammasome activation and signaling. The only well-characterized viral inhibitor of the AIM2 inflammasome is the herpes simplex virus type 1 (HSV-1) protein VP22 that interacts with AIM2, thereby inhibiting the activation of Caspase-1 and processing of IL-1β[22]. VP22 was also required for efficient HSV-1 replication in the brains of intracranially infected mice[22]. However, the impact of VP22 on pyroptosis has not been investigated.

Here we report on the identification of an MCMV inflammasome inhibitor, its ability to inhibit proinflammatory cytokine release and pyroptosis, and its importance for viral replication in cell culture and in its natural host, the mouse.

## Results

### Identification of MCMV M84 as an inhibitor of the AIM2 inflammasome

To identify MCMV proteins that inhibit the AIM2 inflammasome, we co-transfected HEK-293A cells with plasmids encoding proteins of the mouse AIM2 inflammasome together with plasmids from an MCMV ORF library[23], with a focus on known or predicted tegument and related proteins[24,25]. We also tested the non-canonical ORFL147C protein that was identified as an interactor of Caspase-1 in a proteomic screen[26]. As a positive control, we included the HSV-1 protein VP22, a previously described inhibitor of the AIM2 inflammasome[22]. The activation of Caspase-1 was detected by immunoblot analysis (Supplementary Fig. 1a) and the secretion of IL-1β was analyzed by ELISA (Supplementary Fig. 1b). Unfortunately, this transfection-based assay yielded quite variable results with many MCMV ORF expression plasmids producing a moderate degree of inhibition. Therefore, we used a mouse pro-interleukin-1β-Gaussia luciferase (iGLuc) fusion protein[27] as a better quantifiable reporter for the transfection-based screening assay. IL-1β cleavage was detected by measurement of luciferase activity (Fig. 1a). When inflammasome activation was increased by transfecting more ASC expression plasmid, M84 was the most efficient inhibitor (Fig. 1b). These results suggested that MCMV M84 may function as an inhibitor of the AIM2 inflammasome. Interestingly, the AIM2 inhibitor VP22 did not inhibit inflammasome activation when ASC expression was increased (Fig. 1b), probably because elevated ASC levels can lead to ASC oligomerization even in the absence of the sensor[27].

### MCMV M84 interacts with AIM2 and ASC to inhibit inflammasome assembly

The DNA sensor protein AIM2 recruits the adaptor molecule ASC via homotypic interaction of the pyrin domains to form the

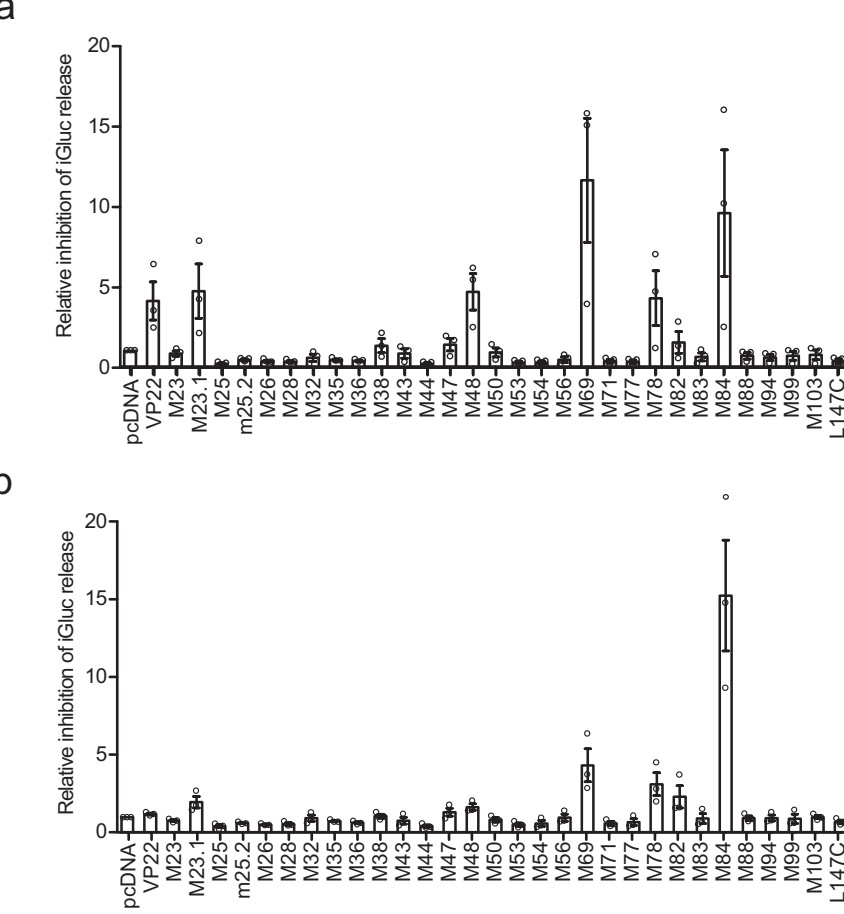

**Fig. 1 | Identification of MCMV M84 as an AIM2 inflammasome inhibitor. a** HEK-293A cells were co-transfected with plasmids encoding mouse AIM2, ASC, Pro-Caspase-1, a Pro-IL-1β-Gaussia luciferase (iGLuc) reporter, and individual MCMV proteins. 24 h post-transfection, supernatants were collected to determine iGLuc release by luciferase assay. The inhibitory effect was calculated by dividing the luciferase reads of vector-transfected cells by those from cells expressing MCMV ORFs. **b** Cells were transfected as described above, but five times more ASC plasmid was used. Mean ± SEM of three independent experiments are shown (**a**, **b**).

inflammasome complex[15]. We speculated that a viral inhibitor of the AIM2 inflammasome might interact with AIM2 and/or ASC. To detect such an interaction, we co-transfected HEK-293A cells with plasmids encoding AIM2 or ASC and MCMV proteins M84, M83, or M82. We chose M82 and M83 as control proteins as they are phylogenetically related to M84 and belong to the same gene family[28]. Flag-tagged AIM2 or ASC were immunoprecipitated from lysates of the transfected cells, and the HA-tagged M82, M83, and M84 proteins were detected by immunoblot analysis. We found that M84, but not M83 or M82, co-precipitated with AIM2 and ASC (Fig. 2a, b). These results suggested that M84 specifically interacts with AIM2 and ASC.

To confirm this finding, we tested whether M84 interacts with AIM2 and ASC during MCMV infection in macrophages. As an M84-specific antibody was not available, we used a recombinant MCMV expressing a C-terminally HA-tagged M84 (Supplementary Fig. 2a). The HA-tagged viral protein was immunoprecipitated from immortalized bone marrow-derived macrophages (iBMDM) infected with MCMV M84HA or MCMV M28HA, which was used as a control. As shown in Fig. 2c, AIM2 and ASC co-precipitated with M84 but not with M28.

Next, we tested whether M84 co-localized with the inflammasome in MCMV-infected macrophages. To do this, we infected primary BMDMs isolated from transgenic mice expressing mCherry-tagged ASC[29] with MCMV M84HA. By immunofluorescence, M84 was detected in the cytoplasm and in the nucleus. In the cytoplasm, M84 co-localized with ASC specks (Fig. 2d).

Next, we used an established flow cytometry-based assay[30] to test whether M84 can inhibit inflammasome assembly. HEK-293A cells were transfected with plasmids encoding AIM2 and GFP-tagged ASC. Cells were co-transfected with M83 or M84 expression plasmids or empty vector, and ASC speck formation was quantified by flow cytometry. Indeed, ASC speck formation was inhibited by M84, but not by M83 (Fig. 2e, f, Supplementary Fig. 2b, c). Taken together, these results indicated that M84 interacts with the inflammasome components AIM2 and ASC and inhibits inflammasome assembly.

## M84 interacts with the pyrin domain

ASC contains a pyrin and a CARD domain, and AIM2 contains a pyrin and a HIN domain and is a member of the PYHIN (IFI200/HIN-200) family of proteins (Fig. 3a). PYHIN proteins are involved in the defense against viral infection[31]. As ASC and AIM2 both contain a pyrin domain (PYD), we hypothesized that M84 interacts with these proteins through the pyrin domain. This hypothesis was confirmed in co-immunoprecipitation experiments. M84 interacted with full-length ASC and the ASC PYD but not with ASC lacking the PYD (ASCΔPYD, Fig. 3b).

We also tested whether M84 interacts with other PYHIN family proteins such as the interferon-inducible protein IFI203 and IFI204 (Fig. 3a). Indeed, M84 co-precipitated with IFI203 and IFI204, but M83 did not (Fig. 3c). These results suggested that M84 can interact with PYHIN family proteins by binding to the PYD.

## M84 inhibits AIM2 inflammasome-mediated restriction of MCMV replication in macrophages

To determine the impact of M84 on MCMV replication, we constructed an MCMV M84stop mutant based on MCMV M84HA by introducing a point mutation in codon 61 of the M84 ORF resulting in a stop codon. The integrity of the viral genomes of MCMV M84HA and M84stop was verified by Illumina sequencing to rule out unintended mutations elsewhere in the viral genome. Then we compared the replication of the two viruses by multistep replication kinetics in 10.1 fibroblasts, J774A.1 macrophages, and iBMDM. In fibroblasts, which lack AIM2 and ASC, the MCMV M84stop mutant replicated with the same kinetics as the parental virus, MCMV M84HA (Fig. 4a). In J774A.1 macrophages and iBMDM, by contrast, the M84stop mutant displayed a clear growth defect (Fig. 4b, c). When macrophages were treated with the Caspase-1-specific inhibitor VX-765, the growth defect was rescued (Fig. 4d).

These results suggested that M84 is required for efficient MCMV replication in AIM2 inflammasome-expressing macrophages but not in fibroblasts. MCMV M84HA replicated to similar titers as WT MCMV in J774A.1 macrophages (Supplementary Fig. 3a), indicating that the C-terminal HA tag of M84 does not negatively affect MCMV replication.

To find out whether the growth defect of the M84stop mutant in macrophages was AIM2 inflammasome-dependent, we generated gene knockouts of Aim2 and Asc in iBMDM by CRISPR/Cas9 gene editing. Two independent cell clones generated with different guide RNAs were tested for each gene. iBMDM transduced with an empty vector (EV, without gRNA) were used as control cells. As shown in Fig. 4e, Caspase-1 activation and GSDMD cleavage were not detectable in Aim2 and Asc knockout iBMDM stimulated with LPS plus poly(dA:dT). Remarkably, the replication defect of M84stop MCMV in WT iBMDM and EV iBMDM (Fig. 4c, Supplementary Fig. 3b) were rescued to WT levels in Aim2 KO (Fig. 4f, Supplementary Fig. 3c) as well as in Asc KO iBMDM (Fig. 4g, Supplementary Fig. 3d). These findings suggested that the AIM2 inflammasome restricts MCMV replication in macrophages and that M84 counteracts this restriction.

As both AIM2 and ASC were involved in impairing the replication of the M84stop mutant, we wanted to know whether the interaction of M84 with these two proteins is interdependent. We tested this with co-immunoprecipitation experiments in AIM2 and ASC-deficient iBMDM. While ASC co-precipitated with M84 in AIM2-deficient iBMDM (Supplementary Fig. 4a), AIM2 co-precipitation with M84 was not detected in ASC-deficient iBMDM (Supplementary Fig. 4b). These results suggested that the interaction of M84 with AIM2 in macrophages is ASC-dependent or that M84 has a higher affinity to ASC than to AIM2.

## M84 inhibits proinflammatory cytokine release and pyroptosis

To find out how M84 affects AIM2 inflammasome signaling upon infection, we infected J774A.1 macrophages with WT MCMV or the M84stop mutant and analyzed the release of Caspase-1 into the supernatant. Upon high-MOI infection with WT MCMV, the fully processed Caspase-1 p20 fragment was detected. However, infection with M84stop resulted in increased p20 levels compared to WT MCMV-infected cells (Fig. 5a).

Upon activation, Caspase-1 releases the cytokines IL-1β and IL-18 from their inactive precursors by proteolytic cleavage. Therefore, we measured IL-18 release from MCMV-infected macrophages by ELISA. As shown in Fig. 5b, infection with M84stop MCMV resulted in significantly higher IL-18 levels compared to infection with WT MCMV. Due to the poor sensitivity of the commercial ELISA assays, we were unable to detect IL-1β released into the supernatant. Therefore, we used a pro-IL-1β-Gaussia luciferase (iGLuc) reporter system, which was designed to overcome this problem[27]. We generated iBMDM stably expressing iGLuc by retroviral transduction and measured luciferase release upon MCMV infection. Again, infection with M84stop significantly increased IL-1β-luciferase release compared to WT MCMV infection (Fig. 5c).

Cleavage of GSDMD is a hallmark of pyroptosis[32]. To test whether M84 can reduce pyroptosis, we analyzed the level of the N-terminal fragment of cleaved GSDMD in J774A.1 macrophages at 10 and 24 h post infection. We found that the M84stop mutant induced higher levels of cleaved GSDMD than WT MCMV (Fig. 5d). To test whether the increased GSDMD cleavage also resulted in a loss of membrane integrity and cell death, we stained infected macrophages with the cell-impermeable dye propidium iodide (PI) and cell-permeable dye Hoechst 33342. PI staining allowed the identification of cells that had lost membrane integrity. Consistent with the results of the GSDMD assay, infection with the MCMV M84stop mutant resulted in more PI-positive cells than infection with WT MCMV, both in J774A.1 macrophages and WT iBMDM (Fig. 5e). Similar results were obtained with an LDH release assay (Supplementary Fig. 5). When the iBMDM were treated with VX-765, there was no difference between WT MCMV and

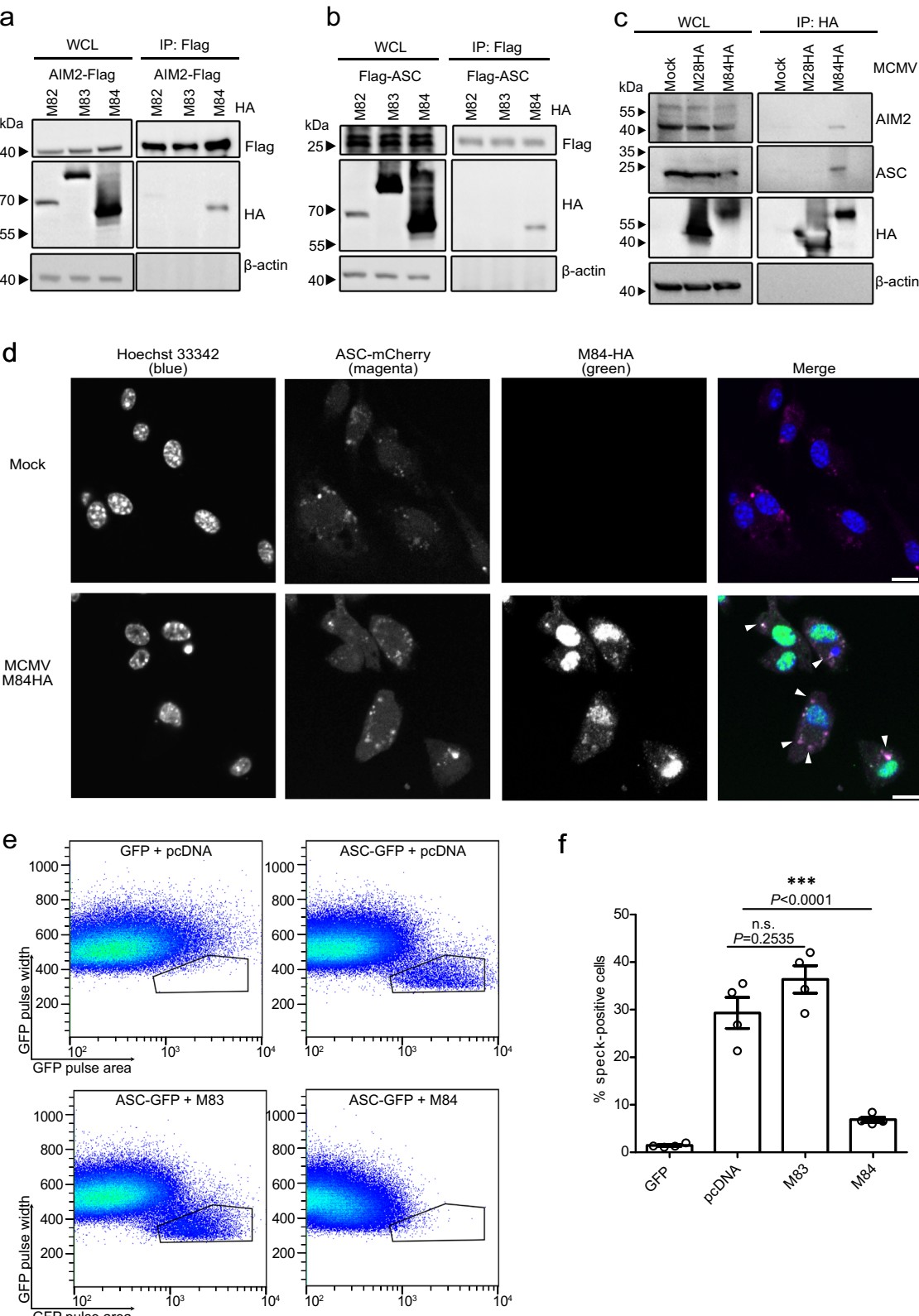

M84*stop*-infected cells (Fig. 5e). Infection of *Aim2* or *Asc* knockout iBMDM resulted in very low levels of PI-positive cells, both with WT MCMV and the M84*stop* mutant (Fig. 5e). Taken together, these results demonstrated that MCMV infection of macrophages activates the AIM2 inflammasome, resulting in Caspase-1 activation, the release of IL-1β and IL-18, GSDMD cleavage, and pyroptosis. In the absence of M84, activation of this signaling cascade was significantly increased,

indicating that M84 inhibits the activation of AIM2 inflammasome and dampens cytokine release and pyroptosis.

## M84 is required for efficient MCMV replication and dissemination in vivo

To analyze the biological impact of M84-mediated inflammasome inhibition on MCMV replication in vivo, we infected WT and

**Fig. 2 | Interaction of M84 with AIM2 and ASC and inhibition of ASC speck formation.** HEK-293A cells were transfected with plasmids encoding AIM2-Flag (**a**) or Flag-ASC (**b**) and HA-tagged MCMV proteins M82, M83 or M84. Whole cell lysates (WCL) were collected 24 h post transfection for immunoprecipitation (IP). Co-precipitating proteins were detected by immunoblot analysis. **c** iBMDM were infected with MCMV M84HA or M28HA (MOI = 5). At 10 hpi, cell lysates were collected for IP using an anti-HA affinity matrix. Co-precipitating proteins were detected by immunoblot analysis with AIM2 and ASC-specific antibodies. **d** Primary BMDM from ASC-mCherry mice were infected with MCMV M84HA (MOI = 3). M84HA was detected with an anti-HA antibody and a secondary antibody coupled

to AlexaFluor488. Nuclei were stained with Hoechst 33342. Micrographs taken by confocal microscopy are shown as monochrome images for better contrast and in color for the merged image. Scale bar, 10 μm. The results are representative of three independent experiments. **e** HEK-293A cells were co-transfected with plasmids encoding AIM2, ASC-GFP or GFP (control), and M83 or M84 or empty vector (pcDNA). ASC speck formation was analyzed by flow cytometry. **f** The percentage of ASC speck-positive cells relative to high GFP-positive cells is shown as mean ± SEM of four independent experiments. Statistical analysis was performed by one-way ANOVA. n.s., not statistically significant; ***, *P* < 0.001. Data are representative of three (**a**–**d**) or four (**e**) biologically independent experiments.

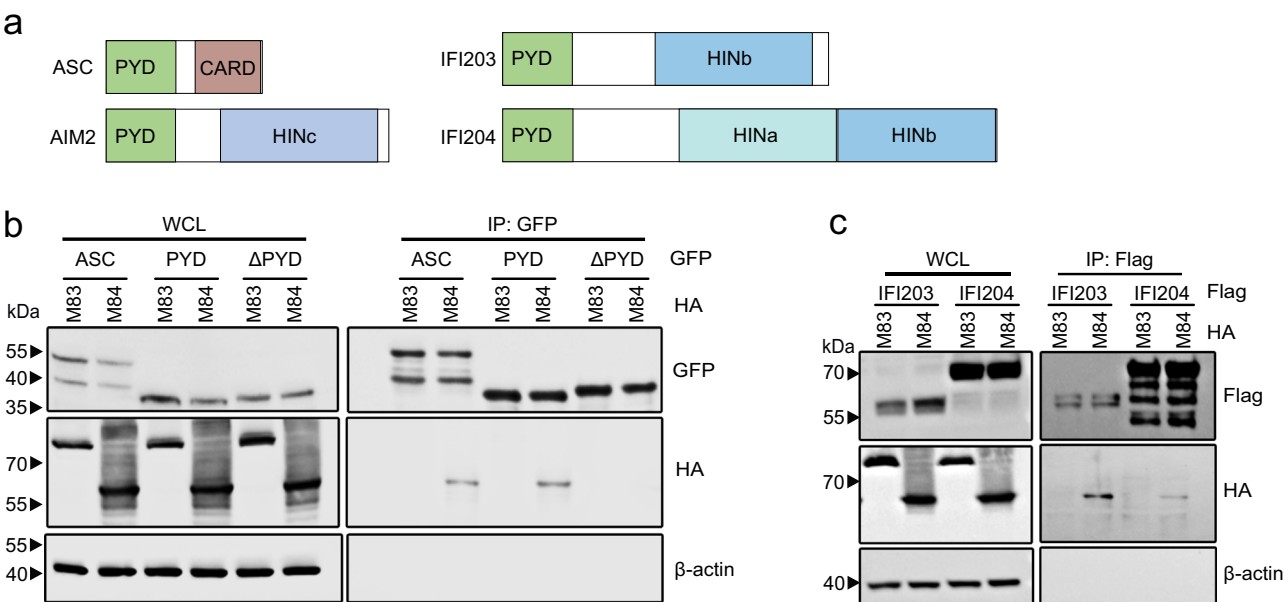

**Fig. 3 | Interaction of M84 with pyrin domain-containing proteins. a** Schematic representation of the domain structures of ASC and PYHIN family proteins AIM2, IFI203, and IFI204. **b** HEK-293A cells were co-transfected with plasmids encoding the GFP-tagged full-length ASC, ASC pyrin domain (PYD), or ASCΔPYD and HA-tagged M83 or M84. **c** HEK-293A cells were transfected with plasmids encoding HA-

tagged MCMV proteins M83 or M84 and IFI203-Flag or IFI204-Flag. Whole-cell lysates (WCL) were collected 24 h post-transfection for immunoprecipitation (IP) and immunoblot analysis (**b** and **c**). The data are representative of three independent experiments.

*Asc*-deficient C57BL/6 mice[33] with WT MCMV and M84*stop* MCMV and determined viral titers in the spleens and livers on day 3 post-infection. As shown in Fig. 6a, b, the viral titers in the spleens of WT mice infected with M84*stop* MCMV were significantly (12-fold) lower than those of mice infected with WT MCMV. In *Asc*-deficient mice, spleen titers of MCMV M84*stop* were only 4-fold lower than those of WT MCMV. Similarly, liver titers of M84*stop* were 17-fold lower than those of WT MCMV in WT mice but only 6-fold lower in *Asc*-deficient mice (Fig. 6c, d). These results showed that MCMV replication in the spleen and liver is impaired in the absence of M84. This defect was partially, but not completely, rescued in *Asc*-deficient mice, suggesting that M84 has additional functions besides inhibition of the AIM2 inflammasome. As M84 also interacts with the PYHIN family protein IFI203 or IFI204, the additional functions might be related to these proteins.

IL-18 is an important cytokine in the early defense against MCMV infection[34]. To investigate the role of M84 in regulating IL-18 levels in vivo, we measured IL-18 levels in the serum of MCMV-infected mice at 1.5 and 3 days post-infection. In WT mice, M84*stop* MCMV induced higher IL-18 serum levels than WT MCMV at 1.5 days post-infection (Fig. 6e). However, on day 3 post-infection, IL-18 serum levels were higher in WT MCMV-infected mice (Fig. 6e), probably due to significantly higher viral loads on day 3 (Figs. 6a and 6c). In *Asc*-deficient mice, IL-18 serum levels were not detectable, consistent with previously published data[19].

To test whether the presence or absence of M84 affects MCMV dissemination, we infected C57BL/6 mice with WT or M84*stop* MCMV at a peripheral site (footpad). Salivary gland titers on day 14 post-infection were significantly reduced in M84*stop*-infected mice (Fig. 6f), suggesting that MCMV dissemination to the salivary gland is impaired in the absence of M84.

Taken together, these data showed that M84 is required for efficient MCMV replication and dissemination in vivo and inhibits the release of the inflammasome-activated cytokine IL-18 in the early stage of MCMV infection in vivo.

## Discussion

CMVs express several cell death suppressors, which inhibit apoptosis and necroptosis, but a CMV inhibitor of pyroptosis, the third major PCD modality, has not been identified (review in ref. 35). Pyroptosis is triggered by inflammasome activation and Caspase-1-mediated cleavage of GSDMD, and the CMVs are known to activate the AIM2 inflammasome[19,36]. We identified the MCMV M84 protein as a viral inflammasome inhibitor and showed that it interacts with AIM2 and ASC, inhibits Caspase-1 activation, release of IL-1β and IL-18, GSDMD cleavage, and pyroptosis. Thus, M84 is a CMV inhibitor of pyroptosis and one of the few viral proteins known to inhibit pyroptosis (reviewed in ref. 2). At present, we do not know whether M84 is the only MCMV inflammasome inhibitor. Our inhibitor screen revealed apparent

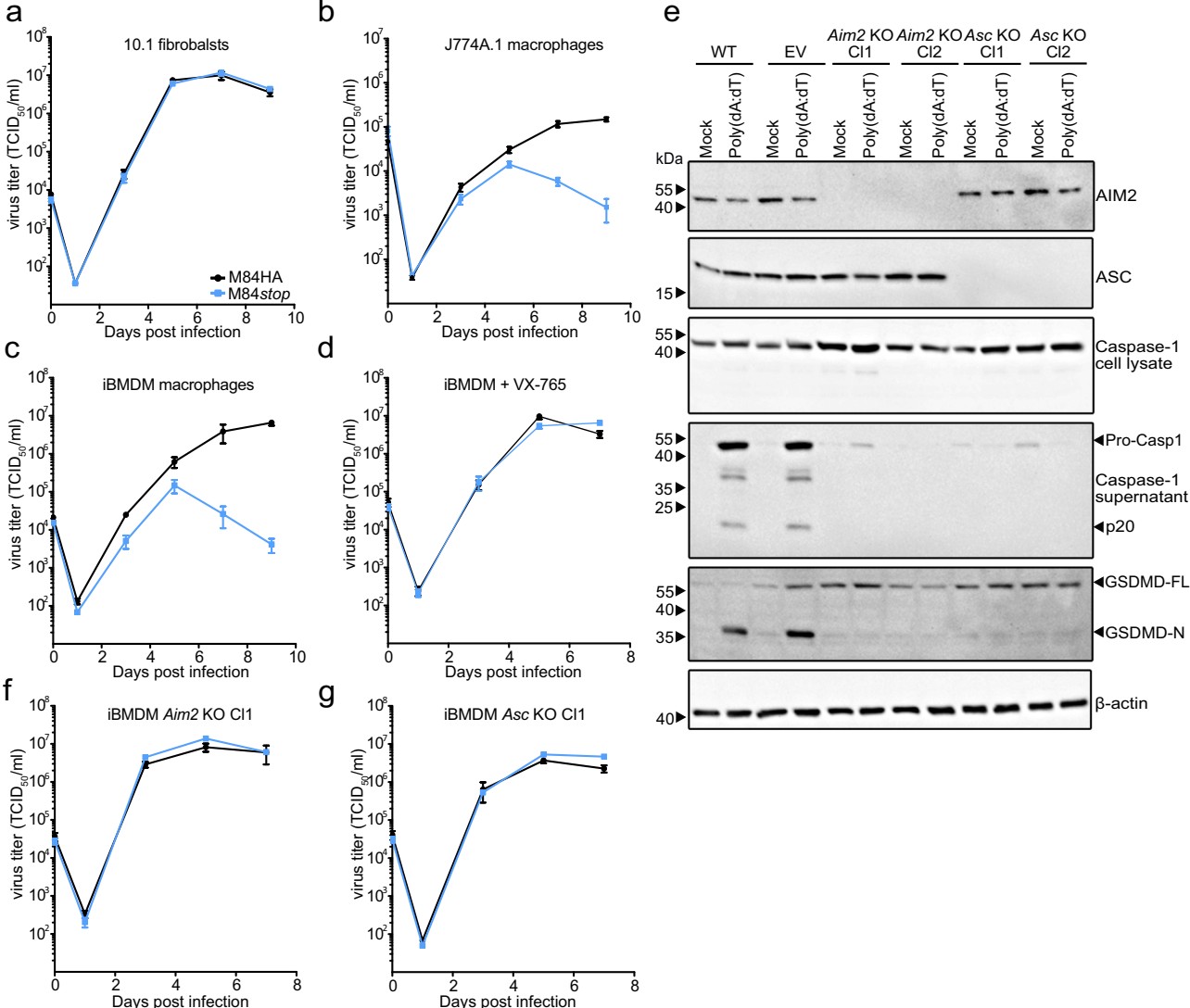

**Fig. 4 | M84 counteracts AIM2 inflammasome-mediated restriction of MCMV replication in macrophages. a** Multistep replication kinetics of MCMV M84HA and M84*stop* in mouse 10.1 fibroblasts, **b** J774A.1 macrophages, **c** immortalized bone marrow-derived macrophages (iBMDM), **d** iBMDM treated with Caspase-1 inhibitor VX-765, **f** *Aim2* KO iBMDM, and **g** *Asc* KO iBMDM. Fibroblasts were infected at MOI = 0.01, J774A.1 at MOI = 0.5, and iBMDM at MOI = 0.025. Viral titers are shown as mean ± SEM of three biological replicates. **e** *Aim2* and *Asc* KO iBMDM were generated by CRISPR/Cas9 gene editing. iBMDM transduced with an empty vector (EV, without gRNA) was used as a control. AIM2 and ASC expression were verified by immunoblot analysis. Caspase-1 activation and GSDMD cleavage in LPS plus Poly (dA:dT) stimulated WT, EV, *Aim2* KO, *Asc* KO iBMDM were also detected by immunoblot analysis. Data are representative of three biologically independent experiments.

inhibitory activities for a few other viral proteins, most notably M69 (Fig. 1). As M69 and its ortholog in HCMV, UL69, have a known function in regulating mRNA export from the nucleus[37], we interpreted this apparent inhibition as an indirect effect. However, we have not further investigated M69 and, therefore, cannot exclude a direct effect of M69 on inflammasome signaling.

Cleavage of GSDMD is a hallmark of pyroptosis[32]. Coronaviruses and picornaviruses express proteins interfering with GSDMD cleavage, thereby preventing pore formation by the GSDMD N-terminal fragment[38–40]. We presumed that a viral inflammasome inhibitor would be a more powerful immune modulator as it would inhibit both pro-inflammatory cytokine release and pyroptosis. Our screen would not have identified an inhibitor specifically targeting GSDMD as the readout of our screen was IL-1β cleavage. Therefore, we used a functional screen that identified the MCMV M84 protein as an inhibitor of AIM2-inflammasome signaling. Our studies showed that MCMV replication in cultured macrophages and dissemination in vivo are impaired in the absence of M84. It is important to note that the significance of a viral immune evasion protein for pathogenesis in vivo can only be investigated if the virus infects laboratory animals (preferably mice) and causes a similar pathology. However, most human herpesviruses (such as HCMV, Epstein-Barr virus, and Kaposi sarcoma-associated herpesvirus [KSHV]) are species-specific and do not replicate in mice[41]. Therefore, related animal viruses have to be used for pathogenesis studies in vivo. MCMV is a natural mouse pathogen and therefore perfectly suited for studies in the mouse model[6].

Two other herpesvirus proteins have been shown to inhibit inflammasome assembly: The KSHV ORF63 protein functions as a viral NLRP1 homolog[42] and the HSV-1 VP22 protein binds to AIM2 and inhibits its oligomerization[22]. Whether these proteins inhibit pyroptosis has not been investigated, but their mechanism of action suggests that they might. As HSV-1 is one of the few human herpesviruses that infect mice, the importance of VP22 could be demonstrated in a mouse model of HSV encephalitis[22]. However, limitations of

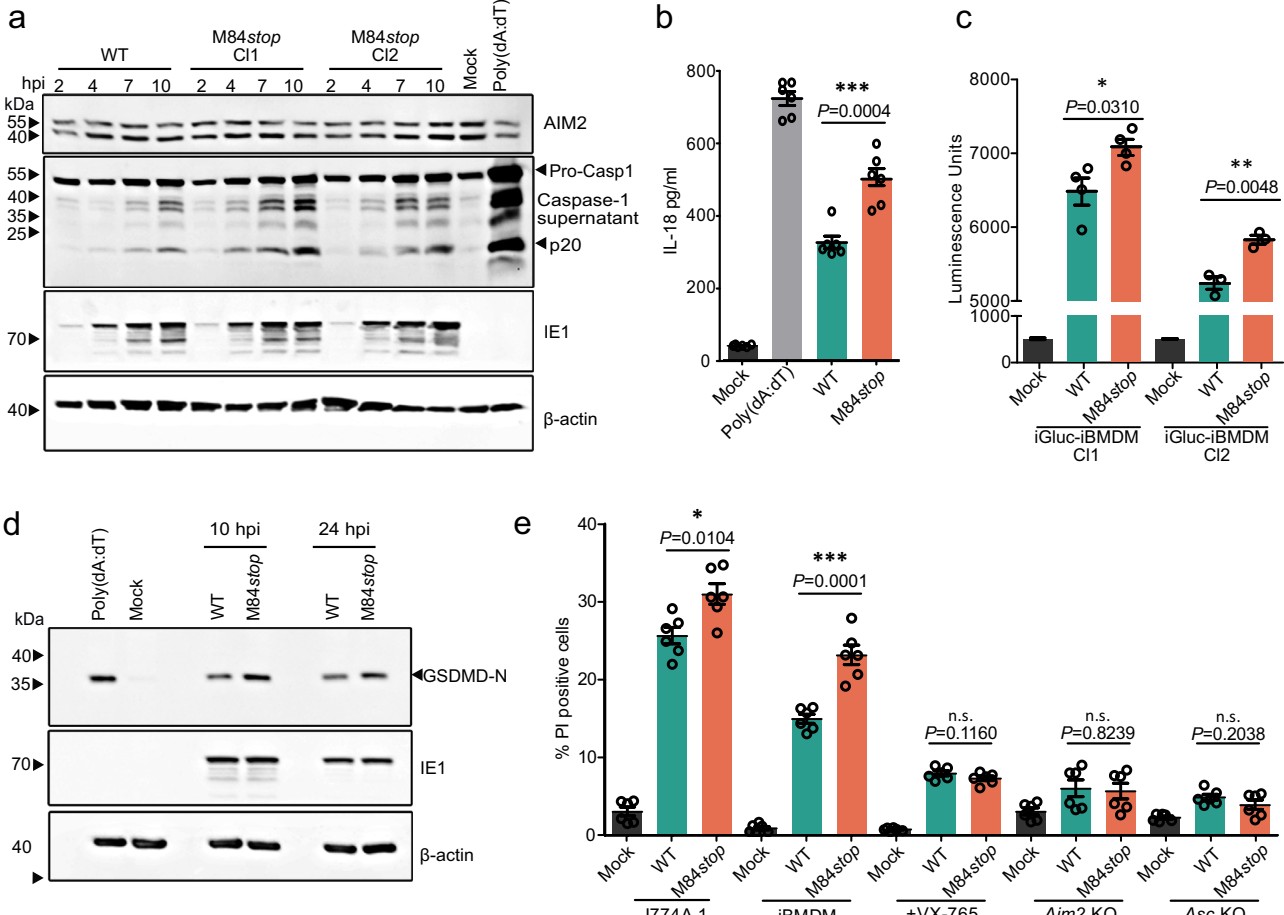

**Fig. 5 | Inhibition of AIM2 inflammasome signaling by M84. a** J774A.1 macrophages were infected with WT MCMV (green) or M84*stop* (orange) (MOI = 5). At the indicated time points, proteins in the supernatant were precipitated and Caspase-1 was analyzed by immunoblot. Mock-infected cells served as negative control and cells treated with LPS plus poly(dA:dT) as positive control. **b** J774A.1 macrophages were infected with WT MCMV or M84*stop* (MOI = 3). The release of IL-18 into the supernatant was analyzed at 7 hpi by ELISA. Mean ± SEM of three independent experiments with two biological replicates each are shown. **c** iBMDM expressing an IL-1β-luciferase reporter were infected with WT MCMV or M84*stop* (MOI = 3). Two transduced iBMDM cell clones (cl1 and cl2) were used. Luciferase activity was measured at 7 hpi and is shown as mean ± SEM of four (cl1) or three (cl2) independents experiments. **d** J774A.1 macrophages were infected with WT MCMV or M84*stop* (MOI = 5). GSDMD cleavage was detected by immunoblotting at 10 and 24 hpi. **e** J774A.1, WT, *Aim2* KO and *Asc* KO iBMDM, and iBMDM treated with VX-765 were infected with WT MCMV or M84*stop* (MOI = 3) and stained at 7 hpi with propidium iodide (PI) and Hoechst 33342. The percentages of PI-positive cells are shown as mean ± SEM of three independent experiments with two biological replicates each are shown. Statistical analysis was done by using the two-tailed Student's *t* test. n.s., not statistically significant; *$P < 0.05$, ** $P < 0.01$, *** $P < 0.001$ (**b**, **c**, and **e**). Data in (**a** and **d**) are representative of three biologically independent experiments.

the HSV-1 mouse infection model must be considered as some HSV-1 immune evasion proteins do not function properly in the mouse. For instance, the HSV-1 ICP6 protein inhibits necroptosis in human cells but, paradoxically, activates necroptosis in mouse cells[43–45].

MCMV infection activates the AIM2 inflammasome and triggers the release of proinflammatory cytokines, as shown here and in a previous study[19]. The viral M84 protein dampens inflammasome activation, cytokine release, and pyroptosis, but does not completely suppress these effects. The reasons for this are unclear. M84 has not been classified as a tegument protein[24] and is therefore not imported into cells at the time of virus entry. In infected macrophages, we detected M84 expression starting at 4 hpi (Supplementary Fig. 2). Hence, it seems unlikely that M84 inhibits AIM2 inflammasome activation by incoming virions. It is conceivable that a certain level of inflammasome activation is advantageous for the virus immediately after infection. Similarly, MCMV first activates and later inhibits NF-κB, a transcription factor driving proinflammatory gene expression[46,47]. At later times, M84 is likely important for inhibiting inflammasome activation by newly synthesized viral genomes and/or by cellular DNA, such as mitochondrial DNA released by damaged mitochondria[48].

The M84-deficient MCMV has a clear growth defect in macrophages, which is completely rescued by a Caspase-1 inhibitor or by knockout of *Aim2* or *Asc*. However, the growth defect is not completely rescued in *Asc* knockout mice in vivo, suggesting that M84 might have additional functions besides inflammasome inhibition. These might be mediated by ALRs IFI203 and IFI204, which also interacted with M84 in co-immunoprecipitation experiments. The functions of these proteins are not fully understood, but all of them can interact with other pyrin domain-containing proteins through homotypic interactions[49]. IFI203 is thought to function as a negative regulator of AIM2, whereas IFI204 was proposed to function as a DNA sensor similar to IFI16 in human cells. Moreover, some of these proteins may also play a role in cell cycle regulation[49]. M84-mediated modulation of these functions would not be rescued by the knockout of *Asc*.

MCMV M84 is a member of the M82-84 gene family, which is homologous to the UL82-84 gene family in HCMV[28]. This raises the question of whether M84 has a functional ortholog in HCMV. UL83 and UL84 are the strongest candidates as they share a similar level of amino acid identity to M84[28]. UL83 is known to interact with IFI16 and inhibit activation of STING[50]. Another study has detected an interaction of

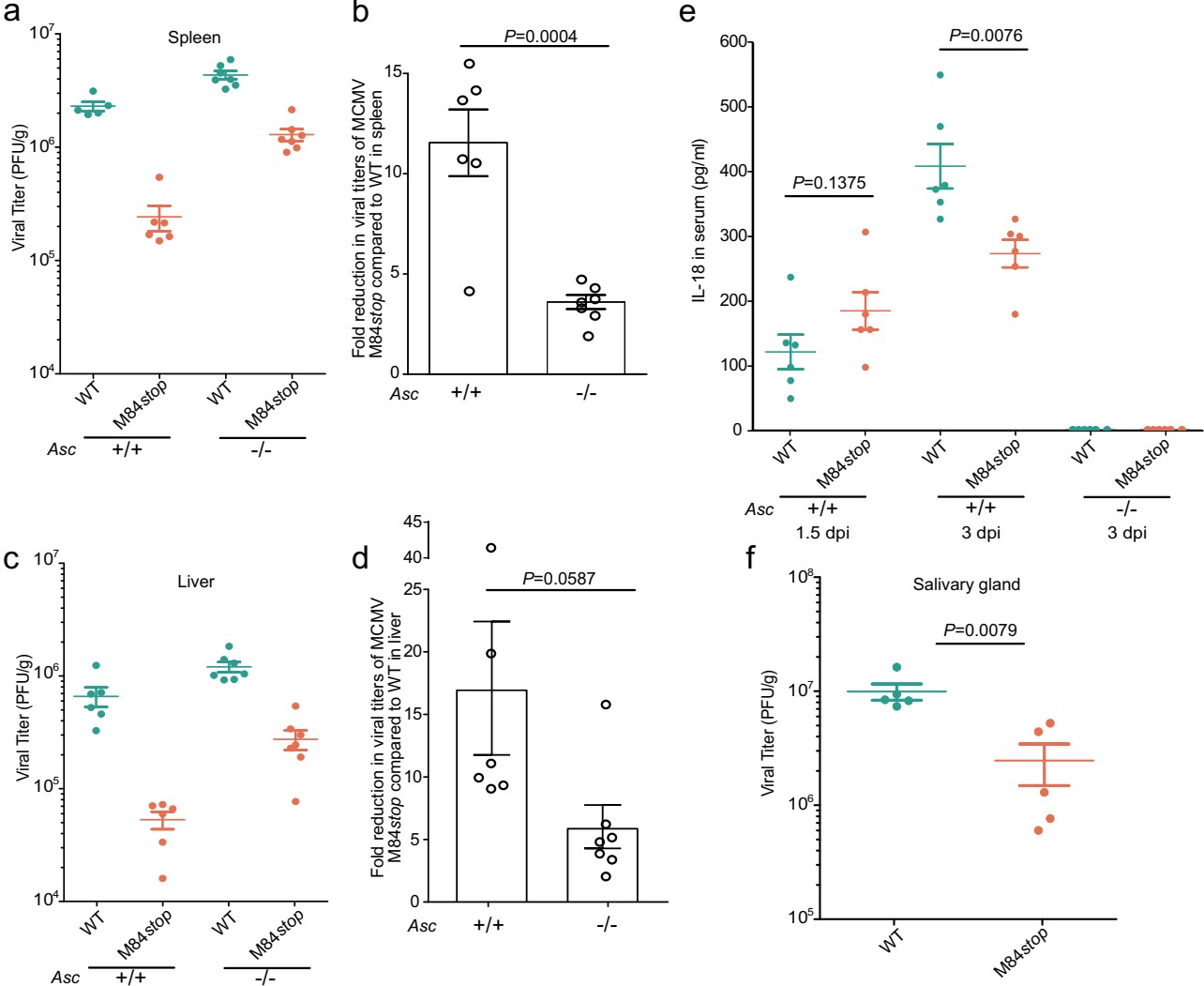

**Fig. 6 | M84 is required for efficient viral replication in vivo.** 6-week-old female WT (*n* = 5 or 6 mice per group) and *Asc*$^{-/-}$ C57BL/6 mice (*n* = 7 mice per group) were infected intraperitoneally with 10$^6$ PFU WT MCMV (green) or M84*stop* (orange). MCMV titers in spleen (**a**) and liver (**c**) determined on day 3 post-infection by plaque assay are shown as mean titers ± SEM. The fold reduction in viral titers of MCMV M84*stop* compared to WT in spleen (**b**) and liver (**d**) of WT or *Asc*$^{-/-}$ mice are shown as mean ± SEM. **e** WT and *Asc*$^{-/-}$ mice (*n* = 6 mice per group) were infected as described above. IL-18 levels in the serum were measured at 1.5 and 3 days post-infection by ELISA. Mean ± SEM is shown. **f** WT C57BL/6 mice (*n* = 5 mice per group) were infected with 10$^5$ PFU WT MCMV or M84*stop* by footpad injection. Viral titers in the salivary glands were determined on day 14 post-infection. Statistical analysis was done by using the two-tailed Student's *t* test (**b, d, e**) or the two-tailed Mann-Whitney test (**f**).

UL83 with AIM2[51], but the consequences of this proposed interaction have not been investigated in virus-infected cells. It is also unknown whether UL83 interacts with ASC as M84 does. Blocking the inflammasome at the level of ASC would not only inhibit the AIM2 inflammasome, but also others such as the NLRP3 inflammasome, which was recently shown to be activated by the cGAS-STING signaling axis in response to cytoplasmic DNA[52,53]. On the other hand, other studies have shown that HCMV activates IL-1β production in human macrophages in an AIM2-dependent manner[36,54]. Whether HCMV infection is sensed by the AIM2 or the NLRP3 inflammasome, and whether UL83 or UL84 inhibits inflammasome activation, cytokine release, and pyroptosis will have to be clarified in future studies.

The AIM2 inflammasome is expressed in hematopoietic cells, particularly those of the myeloid lineage[18], but not in parenchymal and stromal cells of many organs. Myeloid cells, such as monocytes, macrophages, and dendritic cells, play an important role in MCMV dissemination[55–58]. Indeed, the dissemination of MCMV M84*stop* to the salivary glands was reduced upon infection at a peripheral site (Fig. 6f). However, the M84-deficient virus might have replicated in the salivary glands after it reached the organ, and this might account for the moderate titer reduction. Inflammasome inhibition by M84 might also be relevant during reactivation from latency. Myeloid progenitor cells and monocytes are important sites of HCMV latency[59], and latent MCMV infection has also been reported in myeloid cells[60–62]. Therefore, inflammasome inhibition might be advantageous for the virus upon reactivation from latency. It is not known whether MCMV M84 is expressed in latently infected cells, but HCMV UL84 has been detected in latently infected monocytes and CD34+ myeloid progenitor cells[63]. Whether M84 and UL84 can inhibit inflammasome activation during latency and reactivation should be an attractive research question for future studies.

## Methods
### Cells
M2-10B4 (CRL-1972), HEK-293T (CL-11268), and Phoenix-Ampho (CRL-3213) cells were obtained from the American Type Culture Collection. HEK-293A cells (R705-07) were purchased from Invitrogen. Mouse 10.1 fibroblasts have been described[64]. Mouse J774A.1

macrophages (ECACC 91051511) were obtained from the European Collection of Authenticated Cell Cultures and immortalized bone marrow-derived macrophages (iBMDM, NR-9456) from BEI Resources. BMDMs expressing ASC-mCherry were isolated from transgenic mice[29] that were provided by Mathias Gelderblom (University Medical Center Hamburg-Eppendorf). Cells were cultured at 37 °C and 5% $CO_2$ in complete Dulbecco's modified Eagle medium (DMEM) supplemented with 10% fetal calf serum, 100 U/ml penicillin, and 100 μg/ml streptomycin.

## Viruses

WT MCMV (Smith strain) was reconstituted from the full-length BAC clone pSM3fr-MCK-2fl[65]. Mutant MCMVs were generated by *en passant* BAC mutagenesis[66] of pSM3fr-MCK-2fl. MCMV M84HA was made by inserting the HA epitope tag sequence at the 3' end of the M84 ORF. MCMV-M84*stop* was generated by introducing a C-to-G point mutation at the 183rd nucleotide of the M84 ORF leading to a stop codon. MCMV m139HA has been described[67], and MCMV M28HA was constructed in an analogous fashion. To prevent Ly49H-mediated NK cell activation in C57BL/6 mice[68], all viruses used for in vivo experiments were also deleted for m157 as described[69]. To exclude unintended mutations, the mutant BACs were sequence-verified at the NGS facility of the Leibniz Institute of Virology. Infectious virus was reconstituted from BAC DNA by transfection of 10.1 fibroblasts. For stock production, viruses were propagated in 10.1 fibroblasts and the supernatants were collected. Viral particles were pelleted by ultracentrifugation, resuspended, and purified by ultracentrifugation through a sucrose cushion[6]. For in vitro experiments, virus titration was performed on 10.1 cells by using the median tissue culture infective dose ($TCID_{50}$) method. Centrifugal enhancement of infection (30 min at $1000 \times g$) was used for high-MOI infections. For in vivo experiments, virus stocks were titrated on M2-10B4 cells by standard plaque assay[6].

## Antibodies and reagents

Antibodies recognizing the following proteins were used: AIM2 (63660; Cell Signaling), ASC (D2W8U; Cell Signaling), Caspase-1 (AG-20B-0042-C100; AdipoGen), GSDMD (ab209845; Abcam), HA (3F10; Roche), Flag (M2; Sigma-Aldrich), β-actin (AC-74; Sigma), GFP (clones 7.1 and 13.1, Roche). Antibody against MCMV IE1 (CROMA101) was from the Center of Proteomics, University of Rijeka, Rijeka, Croatia. Secondary antibodies coupled to horseradish peroxidase (HRP) were purchased from Jackson ImmunoResearch or DakoCytomation. HRP-coupled anti-Rabbit IgG heavy chain (ab99702) was purchased from Abcam. A secondary antibody coupled to AlexaFluor 488 was purchased from Invitrogen. The caspase-1 inhibitor VX-765 was from Invivogen. All antibodies and dilutions used in this study are listed in Supplementary Table 1.

## Plasmids

An MCMV ORF expression library[23] was kindly provided to us by Niels Lemmermann (University of Mainz, Germany). Plasmids pEPkan-S[66], pSicoR-CRISPR-puroR, pMD2.G, and pCMVR8.91[70] have been described previously. Plasmids encoding mouse inflammasome components were obtained from Addgene: pcDNA3-mAIM2-Flag (plasmid #51537), pcDNA3-N-Flag-ASC (#75134), pCMV-Caspase-1-Flag (#21142), pCMV-pro-IL-1β-c-Flag (#75131). pcDNA3 (Invitrogen), pMSCV-puro (Clontech), pCMV-p203-Myc-DDK (Origene, NM_008328), pCMV-p204-Myc-DDK (Origene, BC010546) were purchased from the indicated sources. ASC-GFP, ASC-PYD-GFP, and ASC-ΔPYD-GFP plasmids were generated by PCR-amplification of the mouse ASC ORF with primers containing HindIII and BamHI restriction sites for cloning in pEGFP-N3. Plasmids encoding HA-tagged M82, M83, and M84 were generated by PCR-amplification of ORFs with primers containing an HA tag sequence and HindIII and EcoRI

restriction sites for cloning in pcDNA3. VP22 was PCR-amplified from HSV-1 and inserted into pcDNA3 with HindIII and BamHI to generate pcDNA-VP22HA. MCMV ORF L147C was PCR-amplified from the MCMV genome and inserted into pcDNA3 with KpnI and EcoRI to generate pcDNA-HAORFL147C. The mouse pro-interleukin-1β-Gaussia luciferase (iGLuc) plasmid pEFBOS-iGLuc[27] was provided by Veit Hornung (University of Munich, Germany). The iGLuc ORF was PCR-amplified and inserted between the BamHI and EcoRI restriction sites of pcDNA3 and the BglII and EcoRI restriction sites of retroviral vector pMSCV-puro.

## Library screening

HEK-293A cells in 12-well plates were transfected with pcDNA3-mAIM2-Flag (200 ng), pcDNA3-N-Flag-ASC (20 or 100 ng), pCMV-Caspase-1-Flag (100 ng), pCMV-pro-IL-1β-c-Flag (200 ng) together with 1 μg plasmid of an MCMV ORF expression library. 24 h post transfection, cells were lysed in 2× SDS-PAGE sample buffer (62.5 mM Tris pH 6.8, 2% SDS, 10% glycerol, 5% β-mercaptoethanol, 0.001% bromophenol blue) for detection of Caspase-1 activation by immunoblot analysis. Cell-free supernatants were collected to determine IL-1β release by ELISA. The inhibitory effect (fold inhibition) was calculated by dividing the IL-1β concentration of empty vector-transfected cells by that of cells transfected with MCMV ORF expression plasmids.

Alternatively, HEK-293A cells were transfected with the same inflammasome plasmids as above, but pcDNA3-iGLuc (200 ng) was used instead of the pro-IL-1β plasmid. The IL-1β-Gaussia luciferase (iGLuc) activity in the supernatant was measured 24 h post transfection.

## Retroviral transduction

iBMDM stably expressing iGLuc[27] were generated by retroviral transduction essentially as described previously[71]. Briefly, Phoenix-Ampho retrovirus packaging cells were transfected with pMSCV-puro-iGLuc. The supernatant was harvested 2 and 3 days post-transfection, passed through a 0.45 μm filter, and used for the transduction of iBMDM. After incubation for 6 h in the presence of 5 μg/mL polybrene (Millipore), the cells were incubated with fresh medium. Transduced cells were selected with 4 μg/mL puromycin (Sigma).

## Generation of gene knockout cells

A lentiviral CRISPR/Cas9 gene editing system[70] was used to generate *Aim2* and *Asc* knockout iBMDM essentially as described[67]. The following gRNAs were designed using E-CRISP (http://www.e-crisp.org/E-CRISP): *Aim2* (gRNA1, GACCACCTGATTCAAAGTGC; gRNA2, GCAGCCTTAGTTCTCAACTC; gRNA3, GACCGGCCTGGACCACATCA) and *Asc* (gRNA1, GTGCAACTGCGAGAAGGCTA; gRNA2, GGACGCTCTTGAAAACTTGT; gRNA3, GCTCAGAGTACAGCCAGAAC). The gRNAs were inserted into the lentiviral vector pSicoR-CRISPR-puroR. Lentiviruses were generated using standard third-generation packaging vectors in HEK-293T cells[70] and were used to transduce iBMDM. Cells were selected with puromycin and single-cell clones were obtained by limiting dilution[67]. Protein expression levels of selected single-cell clones were tested by immunoblot analysis.

## AIM2 inflammasome activation

J774A.1 macrophages or iBMDM were pre-treated with 200 ng/ml ultrapure LPS (Invivogen). The medium was exchanged 3 h later for Opti-MEM (ThermoFisher), and cells were transfected for 3 h with 2 μg/ml poly(dA:dT) (Invivogen) using Lipofectamine 2000 (ThermoFisher). Cell-free supernatants were harvested for the detection of cleaved Caspase-1. Proteins in the supernatant were precipitated as described[72]. Briefly, 500 μL cell-free supernatant was mixed with 125 μL chloroform and 500 μL methanol and centrifuged for 5 min at $13,000 \times g$. The aqueous phase was removed and the lower phase was mixed with methanol and centrifuged for 5 min at $13,000 \times g$. The pellet was dried and resuspended in SDS-PAGE sample buffer. Cell

lysates were collected in SDS-PAGE sample buffer for detection of cleaved GSDMD.

## Immunoprecipitation and immunoblotting

iBMDM were infected with MOI of three in six-well plates for 10 h. Cells were lysed in NP-40 lysis buffer (50 mM Tris, 150 mM NaCl, 1% Nonidet P-40, and Complete Mini protease inhibitor cocktail [Roche]). Following centrifugation to remove the insoluble material, the supernatants were used for immunoprecipitation. HA-tagged proteins was immunoprecipitated with an anti-HA affinity matrix (clone 3F10, Roche). Precipitates were washed three times with buffer 1 (1 mM Tris pH 7.6, 150 mM NaCl, 2 mM EDTA, 0.2% NP-40), twice with buffer 2 (1 mM Tris pH 7.6, 500 mM NaCl, 2 mM EDTA, 0.2% NP-40), once with buffer 3 (10 mM Tris pH 7.6), and then eluted by boiling in SDS-PAGE sample buffer.

HEK-293A cells were transfected with plasmids and lysed 24 h later in RIPA lysis buffer (50 mM Tris-HCl, pH 7.2, 150 mM NaCl, 1% Triton X-100, 0.1% SDS, Benzonase nuclease [Sigma] and Complete Mini protease inhibitor cocktail [Roche]). The GFP-tagged proteins were immunoprecipitated with ChromoTek GFP-Trap Agarose (Proteintech) and Flag-tagged proteins with an anti-Flag antibody (M2, Sigma-Aldrich, 1:500) and protein G Sepharose beads (GE Healthcare). Precipitates were washed 6 times with RIPA buffer and then eluted by boiling in SDS-PAGE sample buffer.

For immunoblot analysis, samples were separated by SDS-PAGE and transferred onto a nitrocellulose membrane (Amersham) by semidry blotting. Blocking was done with 5% nonfat milk diluted in TBST (10 mM Tris, 150 nM NaCl, 1% Tween, pH 7.5). Proteins of interest were detected by enhanced chemiluminescence (GE Healthcare) with specific primary antibodies and HRP-coupled secondary antibodies. Luminescence signals were recorded by using a Fusion Capture Advance FX7 16.15 camera system (Peqlab) or X-ray films.

## Immunofluorescence imaging

BMDM expressing ASC-mCherry were isolated from femurs of transgenic mice[29] as described[69]. Briefly, bone marrow was flushed from femurs and plated for 5 days in DMEM/F12 medium containing 10% FCS and 10% conditioned medium (supernatant of L929 cells containing GM-CSF). Cells were seeded on eight-well μ-slides (Ibidi) and infected on the following day with MCMV-M84HA. Cells were fixed with 4% paraformaldehyde for 15 min at RT. In order to quench free aldehyde groups from fixation, cells were treated with 50 mM ammonium chloride. Cells were permeabilized by incubation in PBS with 0.3% Triton X-100 for 15 min. PBS with 0.2% gelatin (Sigma) was used for blocking and dilution of antibodies. Hoechst 33342 (ThermoFisher) was used to stain nuclear DNA. Fluorescence images were acquired by using a Nikon A1+ confocal laser scanning microscope.

## Detection of cell death

For the detection of pyroptotic cells, macrophages were seeded on 96-well plates and infected on the following day with MCMV (MOI = 3). Cells were stained with 1 μg/ml PI (Sigma) and 5 μg /ml Hoechst 33342. The CellInsight CX5 High-Content Screening Platform (ThermoFisher) was used for automated counting of PI and Hoechst-positive nuclei.

For the detection of LDH release into the supernatant, J774A.1 macrophages or iBMDM were infected with WT MCMV or M84*stop* (MOI = 3). The release of LDH into the supernatant was analyzed at 7 and 10 hpi with a CytoTox 96 Non-Radioactive Cytotoxicity Assay kit (G1780, Promega). A FLUOstar Omega V1.30 plate reader (BMG Labtech) was used to measure the absorbance at 490 nm.

## ASC speck formation assay

Inflammasome reconstitution in HEK-293A cells was done essentially as described[30]. Briefly, HEK-293A cells in 6-well plates were transfected with plasmids encoding mouse AIM2 (500 ng) and ASC-GFP or GFP (500 ng) together with 2.5 μg M83 or M84 or empty vector plasmid.

24 h post-transfection, cells were detached by trypsinization, washed, and fixed with 4% (w/v) paraformaldehyde in PBS at RT for 15 min. Fixed cells were pelleted and resuspended in 700 μL PBS. The number of ASC speck-positive cells was quantified by analyzing ≥10000 GFP+ cells with a FACSCanto II (Beckton Dickinson) flow cytometer. BD FACS Diva (BD Biosciences) was used to acquire flow cytometric data and FlowJo (version 10.8.1; Treestar) was used to analyze flow cytometric data.

## Growth curves

Multistep replication kinetics were done as described[73]. Cells were seeded and infected in 6-well dishes. Three hpi cells were washed twice with phosphate-buffered saline (PBS), and a fresh medium was added. Supernatants harvested from infected cells were titrated on 10.1 cells.

## Cytokine quantification

For measurement of IL-1β, cell-free supernatant was collected 24 h post-transfection. IL-1β production was measured using the mouse IL-1β ELISA kit (Invitrogen, Cat# 88-7013-86) according to the manufacturer's instructions. IL-18 levels in the serum of infected mice and cell-free supernatant of infected macrophages were determined with a Mouse IL-18 ELISA Kit (Invitrogen, Cat# 88-50618-86). The iGLuc reporter in the supernatants of transfected HEK-293A cells and MCMV-infected iBMDM-iGLuc was quantified as described[27] by using Coelenterazine-h (Promega) at a final concentration of 2.5 μM.

## Ethics statement

All animal experiments were performed according to the recommendations and guidelines of the FELASA (Federation for Laboratory Animal Science Associations) and Society of Laboratory Animals (GV-SOLAS) and approved by the institutional review board and local authorities (Behörde für Gesundheit und Verbraucherschutz, Amt für Verbraucherschutz, Freie und Hansestadt Hamburg, reference number N017/2019).

## Animal experiments

C57BL/6 N *Asc*-/- mice (Pycard^tm1Vmd), kindly provided by Vishva Dixit (Genentech), were bred under specific pathogen-free conditions in the small animal facility of the Leibniz Institute of Virology. Mice were maintained on a 12-h light and dark cycle at 21 °C and 55% humidity. Six to eight-week-old female WT or *Asc*-/- C57BL/6 N mice were infected intraperitoneally with $10^6$ PFU MCMV. C57BL/6 J mice (purchased from Janvier Labs) were infected via footpad injection with $10^5$ PFU per mouse. Mice were euthanized by cervical dislocation under isoflurane anesthesia. Spleens and livers were harvested on day 3, and salivary glands on day 14 post-infection. Serum was harvested 1.5 and 3 days post-infection. Viral titers in organs were determined by plaque assay on M2-10B4 cells as described[6]. IL-18 serum concentrations were measured by ELISA as described above.

## Statistical analysis

Statistical analyses were performed using GraphPad Prism 5.0 Software. The two-tailed Student's *t*-test, one-way ANOVA, and two-tailed Mann-Whitney test were used to analyze statistical significance.

## Reporting summary

Further information on research design is available in the Nature Portfolio Reporting Summary linked to this article.

# Data availability

All data generated and analyzed during this study are included in the published article and its supplementary information files or from the corresponding author upon request. Source data are provided with this paper.

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

## Acknowledgements

We thank Mike Munks and Niels Lemmermann for sharing the MCMV ORF library, Vishva Dixit for *Asc*⁻/⁻ mice, Mathias Gelderblom for ASC-mCherry mice, and Veit Hornung for the iGLuc plasmid. We also thank Jana Hennessen and Arne Düsedau from the Flow Cytometry Facility at the Leibniz Institute of Virology for technical support. This work was supported by the Landesforschungsförderung (LFF LV74 to W.B.). The CellInsight CX5 machine was purchased with funds provided by the German Center for Infection Research (DZIF, AD 01.902 to W.B.). Y.D. was supported by a scholarship from the China Scholarship Council (CSC).

## Author contributions

Conceptualization, W.B. and Y.D.; investigation, Y.D. and E.O.; writing—original draft, W.B. and Y.D.; writing—review & editing, all authors; supervision, E.O. and W.B.; funding acquisition, Y.D. and W.B.

## Funding

## Competing interests

The authors declare no competing interests.
