## [Peer Review File · Nature Communications]

A cytomegalovirus inflammasome inhibitor reduces proinflammatory cytokine release and pyroptosisREVIEWER COMMENTS

Reviewer #1 MCMV (Remarks to the Author):

This study by Deng and colleagues shows the exciting result that MCMV encodes a gene to inhibit inflammasome activation. The evidence is convincing that M84 modulates inflammasome activity and can be co-precipitated with inflammasome components. The specific interactions and mechanisms proposed are less well described and several issues should be resolved. In particular, it is very unclear whether M84 is proposed to interact with inflammasome components or the inflammasome complex, how it interacts with different proteins, and how it is thought to inhibit inflammasome function. Specific comments are below.

- 1) IP experiments in Fig 3 show that M84 can interact with PYD domains. However, this experiment is missing the controls of recombinant ASC or AIM2 lacking a PYD domain to confirm that this is the site of interaction and that other sites do not contribute.
- 2) The authors should improve their data showing inflammasome assembly and how M84 interacts with components vs the assembled inflammasome. If the PYD domain is the site of M84 interaction, this raises the question of how the complex between ASC and AIM2 can form in the presence of M84, since these proteins use their PYD domains to oligomerize with each other. The simple explanation would be that M84 prevents inflammasome assembly, but the images shown in Figure 2D seem to suggest that M84 interacts with the ASC-speck, which suggests that M84 does not prevent inflammasome assembly. This is still uncertain however, since “mock” infected cells in Figure 2D appear to show ASC-specks even without infection. This really necessitates showing whether the inflammasome is assembled or not, or if the assembly is reduced in the presence of M84. This can be quantitated by flow cytometry, or by counting specks in cells infected with wt and mutant viruses, or perhaps other ways.
- 3) If M84 is interacting with the PYD domain, but not preventing inflammasome assembly, how is it inhibiting inflammasome function (e.g. Caspase activation)?
- 4) Co-IP experiments indicate both AIM2 and ASC are co-precipitated with M84 in transfected 293A cells. But the data in Figure S3 suggest that M84 did not co-precipitate with AIM2 in ASC-KO cells. This would suggest that M84 is interacting with ASC directly and

AIM2 is being pulled down as part of the complex. So: 1) do the 293A cells express any endogenous AIM2 or ASC? This could explain the pulldown of AIM2 in their earlier co-IP experiments. It would be nice to see whether there was some AIM2 in these cells or in precipitated complexes with ASC if possible. 2) Are the PYD domains very homologous? If so, how is M84 selectively binding to ASC? Are there differences in the ability of M84 to interact with the ASC-PYD domain vs the AIM2-PYD domain?

5) Figure 4F and 4G data seems to be not mentioned in the text and there appears to be no corresponding legend.

6) The cell death increase that is described in Figure 5 is fairly subtle in the absence of M84 and should be verified with a second method (for example, to rule out PI-related toxicity). LDH release is commonly used. Likewise, images of PI-staining are often included, but probably won't look much different when comparing 25% and 30% PI+ cells.

7) The authors describe the in vivo viral growth results in Figure 6 as "impaired dissemination to the spleen and liver in the absence of M84" (line 281). Since this experiment used an i.p. infection, I'm not sure I agree with the term dissemination in this context. Rather, since i.p. infections allow the virus access to the blood stream, this result suggests that loss of M84 inhibited primary replication in these organs. To test dissemination, the authors would have to restrict the primary inoculum via footpad or intranasal infections.

Reviewer #2 MCMV immune evasion (Remarks to the Author):

Deng and colleagues presented in their manuscript the first evidence that the murine cytomegalovirus (MCMV) encodes an inhibitor of the AIM2 inflammasome. They identified the viral protein M84 and showed that it interacts with ASC (in transfection and infection systems) and AIM2 (in transfection system) through their pyrin domains. The authors also showed that M84 also interacts with other pyrin domain-containing proteins - although they did not explore the functional consequences of these interactions. The author demonstrated that, when compared to a wt-virus, growth of an M84-deleted virus is attenuated in wild-type macrophages (but not in ASC or AIM2 -deficient macrophages) and is associated with more GSDMD cleavage, Caspase 1 activation, IL18 and IL1b secretion and cell death. Finally they showed that the M84-deleted virus is attenuated in vivo in mice, in a

partial ASC dependent manner.

Taken together the authors presented in their manuscript an interesting finding that is highly relevant to the fields of virology and immunology. As presented, the paper has a few technical problems that need to be solved or discussed.

The activation of the AIM2 inflammasome is initiated by cytosolic double-stranded DNA (dsDNA) and is thus presumed to take place early during herpesvirus infection. Considering this, it is improbable that the viral protein M84, unlike the tegument proteins VP22 and ORF63, can hinder this activation process. Have the authors considered the possibility that AIM2 activation in cells infected with MCMV may not solely be restricted to the early stages of infection and that other events, occurring during later stages of infection, might also trigger AIM2 activation? For instance, one potential mechanism could involve the release of DNA from damaged mitochondria. Although it is not within the scope of this manuscript to determine whether this is indeed the case, this should be discussed.

The significance and relevance of this study would be enhanced by providing a more comprehensive understanding of the impact of AIM2 inflammasome inhibition on MCMV pathogenesis. For example:

How are viral titers affected at the peak of the infection?

At 3 days post-infection, as presented in this study, both stromal cells (lacking AIM2) and hematopoietic cells (expressing AIM2) are infected and contribute to the overall viral load.

Could it be that the importance of AIM2 inhibition is underestimated in this context?

What is the importance of AIM2 inhibition on latency and reactivation, considering their occurrence in hematopoietic cells?

I acknowledge the challenges associated with these investigations and only wish for a discussion on that topic. This would provide a more comprehensive perspective on the potential role and importance of AIM2 in MCMV pathogenesis.

The origin of the proteins encoding the various components of the AIM2 inflammasome (shown in figure 1) is not specified clearly in the text or the materials and methods section.

Could the authors confirm that all these proteins are from mouse origin. If this is not the case, there is a possibility that additional viral proteins might inhibit the AIM2

inflammasome, but were overlooked due to a lack of interaction in this assay.

It is intriguing that VP22, employed as a positive control in figure 1, demonstrates only a relatively modest inhibitory effect in figure 1A and none in figure 1B. Could the authors provide an explanation for this observation.

In figure 1A, S1A, and S1B, M69 appears to exhibit an inhibitory effect comparable to M84 (only in figure 1B, M69 shows a slightly lower inhibitory effect when compared to M84). What specific experiments have been conducted to establish with confidence that M69 does not function as an additional inhibitor of the inflammasome?

In Figure 5A, the presence of pro-caspase1 in the supernatant is observed, even in cells that were treated with MOCK (where no cell death is anticipated). Can the authors provide an explanation for this unexpected finding? For example, this is not observed in similar assay (figure 4E)

Figure 5B, C, and E demonstrate that infection with the M84stop virus leads to a higher induction of IL18 secretion, luciferase activity, and cell death compared to the wild-type virus. Although these differences are statistically significant, they are not substantial. Have the authors investigated the expression levels of M84 in infected cells, using for example their HA-M84 virus? A western blot showing the expression kinetic of M84 following infection should be provided.

In Figure 5D, the alignment of lanes in the β -actin blot does not correspond to the other blots (as well as the legend). Is this due to a technical issue? The same comment applies to Figure S3.

Reviewer #3 CMV, immune interaction, inflammasome (Remarks to the Author):

This manuscript does a very complete substantiation of the role MCMV M84-encoded protein plays in modulating AIM2 inflammasome activation. The data are complete and clearly presented.

Point-by-Point Response to the Reviewer Comments

We would like to thank all three reviewers for taking the time to read and evaluate our manuscript. Their insightful and supportive comments helped us improve the manuscript.

Reviewer #1 MCMV (Remarks to the Author):

This study by Deng and colleagues shows the exciting result that MCMV encodes a gene to inhibit inflammasome activation. The evidence is convincing that M84 modulates inflammasome activity and can be co-precipitated with inflammasome components. The specific interactions and mechanisms proposed are less well described and several issues should be resolved. In particular, it is very unclear whether M84 is proposed to interact with inflammasome components or the inflammasome complex, how it interacts with different proteins, and how it is thought to inhibit inflammasome function. Specific comments are below.

1) IP experiments in Fig 3 show that M84 can interact with PYD domains. However, this experiment is missing the controls of recombinant ASC or AIM2 lacking a PYD domain to confirm that this is the site of interaction and that other sites do not contribute.

Response: Thank you for this comment. We agree that the suggested control would strengthen the conclusion. Therefore, we now compared full-length ASC with ASC PYD and ASC lacking PYD (ASC Δ PYD). The results show that M84 interacts with full-length ASC and with the ASC PYD but not with the ASC CARD (Δ PYD). M83 did not interact with ASC. The results are shown in the new Fig. 3b.

2) The authors should improve their data showing inflammasome assembly and how M84 interacts with components vs the assembled inflammasome. If the PYD domain is the site of M84 interaction, this raises the question of how the complex between ASC and AIM2 can form in the presence of M84, since these proteins use their PYD domains to oligomerize with each other. The simple explanation would be that M84 prevents inflammasome assembly, but the images shown in Figure 2D seem to suggest that M84 interacts with the ASC-speck, which suggests that M84 does not prevent inflammasome assembly. This is still uncertain however, since “mock” infected cells in Figure 2D appear to show ASC-specks even without infection. This really necessitates showing whether the inflammasome is assembled or not, or if the assembly is reduced in the presence of M84. This can be quantitated by flow cytometry, or by counting specks in cells infected with wt and mutant viruses, or perhaps other ways.

Response: This is a very good question. The fact that M84 interacts with pyrin domains suggests that M84 probably inhibits inflammasome assembly. We agree that this should be confirmed experimentally. As suggested by this reviewer, we used flow cytometry to measure ASC speck formation. We used a well-described and frequently used protocol (Sester et al., 2015; PMID 25404358) involving transfection of cells with AIM2 and ASC-GFP expression plasmids. Cells were co-transfected with M83 or M84 expression plasmids or empty vector. As shown in the new Fig 2e, 2f and Fig S2c, M84 inhibited ASC speck formation while M83 did not. These results support the conclusion that M84 inhibits inflammasome assembly.

3) If M84 is interacting with the PYD domain, but not preventing inflammasome assembly, how is it inhibiting inflammasome function (e.g. Caspase activation)?

Response: As shown above (response to point #2), M84 does inhibit inflammasome assembly.

4) Co-IP experiments indicate both AIM2 and ASC are co-precipitated with M84 in transfected 293A cells. But the data in Figure S3 suggest that M84 did not co-precipitate with AIM2 in ASC-KO cells.

This would suggest that M84 is interacting with ASC directly and AIM2 is being pulled down as part of the complex. So: 1) do the 293A cells express any endogenous AIM2 or ASC? This could explain the pulldown of AIM2 in their earlier co-IP experiments. It would be nice to see whether there was some AIM2 in these cells or in precipitated complexes with ASC if possible. 2) Are the PYD domains very homologous? If so, how is M84 selectively binding to ASC? Are there differences in the ability of M84 to interact with the ASC-PYD domain vs the AIM2-PYD domain?

Response: This reviewer raises some interesting questions. Co-IP experiments are useful to detect interactions, but they cannot reliably discriminate direct from indirect interactions. 293 cells were previously reported to be AIM2 and ASC negative (Fernandes-Alnemri et al., 2009; PMID 19158676 and Chen et al., 2018; PMID 30487600), and we did not detect endogenous AIM2 or ASC in our 293A cells. An alignment of the pyrin domains of ASC, AIM2 (murine and human), p203 and p204 shows their homology, but the sequence identity/similarity is not very high (see below). However, the structural similarity may be much higher. We did not claim that M84 selectively binds ASC, we merely stated that the inability to co-precipitate AIM2 in ASC knockout cells might indicate an indirect interaction. Another explanation, as suggested by this reviewer, would be differences in the ability of M84 to interact with the ASC-PYD domain vs the AIM2-PYD domain. We mentioned this alternative explanation in the revised manuscript (lines 172 to 174). Determining whether the interaction of M84 with ASC is stronger than its interaction with AIM2, p203, and p204 is a difficult task, which would require a larger set of competition experiments. We believe that such experiments would be interesting, but they would add little to the story presented here and are therefore beyond the scope of this manuscript.

```

1                                     94
mASC  MGRARDAI LDALENLSGD ELKKFKMKLL TVQLREGYGR IPRGALLQMD AIDLTDKLVSY YLESYGLEL TMTVLRDMG- LQELAEQLQT TKEE
hASC  MGRARDAI LDALENLTAE ELKKFKLKL SVPLREGYGR IPRGALLSMD ALDLDKLVSY FLETYGAEL TANVLRDMG- LQEMAGQLQA ATHQ
mAIM2 MESEYREMLL LTGLDHITTE ELKRFKYFAL TEF-----Q IARSTLDVAD RTELADHLIQ SAGAASAVTK AINIFQKLN Y MH-IANALEE KKKE
hAIM2 MESKYKEILL LTGLDNITDE ELDRFKFFLS DEF-----N IATGKHLTAN RIQVATLMIQ NAGAVSAVMK TIRIFQKLN Y ML-LAKRLQE EKKE
p203  MAEYKNIVL LKGLNEMEDY QFRTVKSLLR KEL-----K LTKKMQEDYD RIQLADWMEK KFPKDAGLDK LIKVCEHIKD LKDLAKKLLKT EKAK
p204  MVNEYKRIVL LRGLECIINKH YFSLFKSLLA RDL-----N LERDNQEYQT TIQIANMEE KFPADSGLGK LIAFCEEVPA LRKRAEILKK ERSE
Consensus  ...eykri.l L.gL#ni... e1..fk..l. .e..... i.r..l...d ri#lad.$.. ...a.sg1.k .i.v..... $.lA..L.. ek.e

```

5) Figure 4F and 4G data seems to be not mentioned in the text and there appears to be no corresponding legend.

Response: The reviewer is right, and we apologize for this oversight. We corrected the text and the legend to Fig 4 in the revised version.

6) The cell death increase that is described in Figure 5 is fairly subtle in the absence of M84 and should be verified with a second method (for example, to rule out PI-related toxicity). LDH release is commonly used. Likewise, images of PI-staining are often included, but probably won't look much different when comparing 25% and 30% PI+ cells.

Response: Following this reviewer's advice, we used an LDH release assay for confirmation. Infection of macrophages with the M84stop virus resulted in significantly higher LDH release than WT MCMV infection (new Fig S5), consistent with the results of the PI staining assay.

7) The authors describe the in vivo viral growth results in Figure 6 as "impaired dissemination to the spleen and liver in the absence of M84" (line 281). Since this experiment used an i.p. infection, I'm not sure I agree with the term dissemination in this context. Rather, since i.p. infections allow the virus access to the blood stream, this result suggests that loss of M84 inhibited primary replication in these organs. To test dissemination, the authors would have to restrict the primary inoculum via footpad or intranasal infections.

Response: We agree that it is difficult to discriminate between replication within an organ and dissemination to an organ. Therefore, we toned down statements regarding dissemination (lines 214 ,216 and 223 in the revised manuscript). We also followed this reviewer's suggestion and infected mice via footpad injection and measured dissemination to the salivary gland. Indeed, the M84^{stop} virus reached lower titers in the salivary glands than WT MCMV (new Fig 6f).

Reviewer #2 MCMV immune evasion (Remarks to the Author):

Deng and colleagues presented in their manuscript the first evidence that the murine cytomegalovirus (MCMV) encodes an inhibitor of the AIM2 inflammasome. They identified the viral protein M84 and showed that it interacts with ASC (in transfection and infection systems) and AIM2 (in transfection system) through their pyrin domains. The authors also showed that M84 also interacts with other pyrin domain-containing proteins - although they did not explore the functional consequences of these interactions. The author demonstrated that, when compared to a wt-virus, growth of an M84-deleted virus is attenuated in wild-type macrophages (but not in ASC or AIM2 - deficient macrophages) and is associated with more GSDMD cleavage, Caspase 1 activation, IL18 and IL1b secretion and cell death . Finally they showed that the M84-deleted virus is attenuated in vivo in mice, in a partial ASC dependent manner.

Taken together the authors presented in their manuscript an interesting finding that is highly relevant to the fields of virology and immunology. As presented, the paper has a few technical problems that need to be solved or discussed.

Response: We thank this reviewer for the supportive and constructive comments. As requested, we have solved or discussed all issues (described in the specific responses below).

1. The activation of the AIM2 inflammasome is initiated by cytosolic double-stranded DNA (dsDNA) and is thus presumed to take place early during herpesvirus infection. Considering this, it is improbable that the viral protein M84, unlike the tegument proteins VP22 and ORF63, can hinder this activation process. Have the authors considered the possibility that AIM2 activation in cells infected with MCMV may not solely be restricted to the early stages of infection and that other events, occurring during later stages of infection, might also trigger AIM2 activation? For instance, one potential mechanism could involve the release of DNA from damaged mitochondria. Although it is not within the scope of this manuscript to determine whether this is indeed the case, this should be discussed.

Response: We agree. The expression kinetics of M84 starting around 4 hours post-infection (see response to point #7 below) suggests that M84 inhibits AIM2 inflammasome activation at early and late stages of infection. AIM2 inflammasome activation may be activated by viral DNA or by DNA of cellular origin, such as DNA released from mitochondria. As suggested, we discussed these important considerations in the revised manuscript (lines 297 to 299).

2. The significance and relevance of this study would be enhanced by providing a more comprehensive understanding of the impact of AIM2 inflammasome inhibition on MCMV pathogenesis. For example:

How are viral titers affected at the peak of the infection? At 3 days post-infection, as presented in this study, both stromal cells (lacking AIM2) and hematopoietic cells (expressing AIM2) are infected and contribute to the overall viral load. Could it be that the importance of AIM2 inhibition is underestimated in this context? What is the importance of AIM2 inhibition on latency and reactivation, considering their occurrence in hematopoietic cells?

I acknowledge the challenges associated with these investigations and only wish for a discussion on that topic. This would provide a more comprehensive perspective on the potential role and importance of AIM2 in MCMV pathogenesis.

Response: This reviewer raises a number of interesting questions, and we agree that these should be discussed. In the revised manuscript we added a new paragraph (lines 327 to 341) in which we discuss the presence of the AIM2 inflammasome in hematopoietic cells and its absence in stromal cells and the implications for MCMV replication as well as latency and reactivation in its host. We also include new experimental data showing lower salivary gland titers of the M84*stop* mutant after footpad infection (see also response to reviewer #1, point #7).

3. The origin of the proteins encoding the various components of the AIM2 inflammasome (shown in figure 1) is not specified clearly in the text or the materials and methods section. Could the authors confirm that all these proteins are from mouse origin. If this is not the case, there is a possibility that additional viral proteins might inhibit the AIM2 inflammasome, but were overlooked due to a lack of interaction in this assay.

Response: All plasmids encoding inflammasome components (AIM2, ASC, Caspase-1) are of mouse origin. The plasmids were obtained from Addgene, and the species origin can be verified on the Addgene website. We fully agree that it is important to work with mouse proteins when studying a mouse virus. Therefore, we included this important information in the Results (line 75) and the Materials and Methods (line 396) sections of the revised manuscript.

4. It is intriguing that VP22, employed as a positive control in figure 1, demonstrates only a relatively modest inhibitory effect in figure 1A and none in figure 1B. Could the authors provide an explanation for this observation.

Response: We agree, and we have noted this ourselves. The most likely explanation for the modest inhibition in Fig. 1a is that VP22 is from HSV-1, a human herpesvirus has evolved to inhibit human AIM2. Our assay was done with inflammasome components of mouse origin. The lack of inhibition by VP22 in Fig 1b is most probably due to the different setup. In Fig 1b, we used 5 times more ASC than in Fig 1a. Higher ASC levels can cause ASC oligomerization without activation of the sensor (Bartok et al., 2013, PMID 23291722). VP22 was reported to inhibit AIM2 oligomerization (Maruzuru et al., 2018; PMID 29447697). Apparently, it does not inhibit oligomerization of murine ASC under the conditions used here. We commented on the lack of inhibition by VP22 in Fig 1b in the revised manuscript (lines 89 to 92).

5. In figure 1A, S1A, and S1B, M69 appears to exhibit an inhibitory effect comparable to M84 (only in figure 1B, M69 shows a slightly lower inhibitory effect when compared to M84). What specific experiments have been conducted to establish with confidence that M69 does not function as an additional inhibitor of the inflammasome?

Response: This reviewer raises a valid question. We focused on the strongest inhibitor, M84, and did not further investigate M69. Therefore, we cannot exclude a role of M69 as an additional inhibitor or cofactor. The known functions of M69 and its HCMV ortholog, UL69, in mRNA export (Zielke et al., 2012; PMID 21147923) suggested to us that the observed effect might be an indirect one. However, specific experiments would be necessary to determine whether or not M69 has a direct role in inflammasome inhibition. We discussed the possibility that other viral proteins, besides M84, might be involved in inflammasome inhibition (lines 253 to 259).

6. In Figure 5A, the presence of pro-caspase1 in the supernatant is observed, even in cells that were treated with MOCK (where no cell death is anticipated). Can the authors provide an explanation for this unexpected finding? For example, this is not observed in similar assay (figure 4E)

Response: This observation is correct. The results in Fig 5a were obtained with J774A.1 macrophages whereas the results in Fig 4e were obtained with iBMDM. We regularly observed more spontaneous cell death with J774A.1 cells than with iBMDM. This can also be seen in Fig 5e (compare mock-infected J774A.1 cells and iBMDM). The figure also shows that J774A.1 cells are also more sensitive to cell death induced by high-MOI MCMV infection. However, as the readout is cleaved caspase-1 (p20), not pro-caspase-1, the interpretation is not affected.

7. Figure 5B, C, and E demonstrate that infection with the M84stop virus leads to a higher induction of IL18 secretion, luciferase activity, and cell death compared to the wild-type virus. Although these differences are statistically significant, they are not substantial. Have the authors investigated the expression levels of M84 in infected cells, using for example their HA-M84 virus? A western blot showing the expression kinetic of M84 following infection should be provided.

Response: Thank you for this suggestion. We agree that M84 expression kinetics by Western blot would be informative. We included such an experiment as a new Fig S2a. It shows that M84 expression becomes detectable at 4 hours post-infection in iBMDM.

8. In Figure 5D, the alignment of lanes in the β -actin blot does not correspond to the other blots (as well as the legend). Is this due to a technical issue? The same comment applies to Figure S3.

Response: We apologize for these accidental mistakes that happened while resizing and cropping the blots. The correctly sized and cropped blots are shown in the revised version. This can be verified using the raw data (uncropped blots) submitted along with the revised version.

Reviewer #3 CMV, immune interaction, inflammasome (Remarks to the Author):

This manuscript does a very complete substantiation of the role MCMV M84-encoded protein plays in modulating AIM2 inflammasome activation. The data are complete and clearly presented.

Response: We thank this reviewer for his/her supportive comments.

REVIEWERS' COMMENTS

Reviewer #1 (Remarks to the Author):

The authors have addressed all of my comments and added significant new data. I have no further concerns. It is an excellent study.

Reviewer #2 (Remarks to the Author):

The authors answered all my comments in this revision.

Reviewer #3 (Remarks to the Author):

Authors have provided a thoughtful response to all reviewers' comments. They have provided new data to affirm mechanisms of M84 subversion of inflammasome formation, evidence that enhances the substantive conclusions that CMV encodes a dedicated inflammasome inhibitor. They added other data where the shortcoming was critical to their conclusions. They have provided rationale for not pursuing more expansive studies into less critical, but fascinating questions that will be held for future experimentation. The work is significant, complete and convincing.